# Power Management Unit for Solar Energy Harvester Assisted Batteryless Wireless Sensor Node

**DOI:** 10.3390/s22207908

**Published:** 2022-10-18

**Authors:** Alberto Lopez-Gasso, Andoni Beriain, Hector Solar, Roc Berenguer

**Affiliations:** Department of Electrical and Electronic Engineering, TECNUN—Technological Campus of the University of Navarra, 20018 San Sebastian, Spain

**Keywords:** solar, energy harvesting, power management unit, charge pump

## Abstract

This work describes an energy-efficient monolithic Power Management Unit (PMU) that includes a charge pump adapted to photovoltaic cells with the capability of charging a large supply capacitor and managing the stored energy efficiently to provide the required supply voltage and power to low energy consumption wireless sensor nodes such as RFID sensor tags. The proposed system starts-up self-sufficiently with a light source luminosity equal to or higher than 500 lux using only a 1.42 cm^2^ solar cell and integrating an energy monitor that gives the ability to supply autonomous sensor nodes with discontinuous operation modes. The system occupies an area of 0.97 mm^2^ with a standard 180 nm CMOS technology. The half-floating architecture avoids losses of charging the top/button plate of the stray capacitors in each clock cycle. Measurements’ results on a fabricated IC exhibit an efficiency above 60% delivering 13.14 μW over 1.8 V. The harvested energy is enough to reach the communication range of a standard UHF RFID sensor tag up to 21 m.

## 1. Introduction

Autonomous sensor nodes are the key element for spreading IoT technology and are becoming increasingly popular. The global market is expected to grow annually up to 25.4% in the 2021–2028 period, from 381.30 billion dollars in 2021 to 1854.76 billion dollars in 2028 [1]. Considering the large number of different communication protocols that can be adapted to every case, the next big challenge in the autonomous sensor nodes field is the battery life.

As devices get smaller, powering them with traditional batteries becomes an issue because of size, environmental impact, and maintenance costs. Moreover, replacing batteries periodically, even every 5 or 10 years, is unsuitable for many applications, while photovoltaic cells are fabricated with more than 25 years of lifespan maintaining more than 80% of the efficiency [2]. In addition, facts like unforeseen battery degradation, natural ageing, or instantaneously burnout in chemical rechargeable batteries limit their utilization and are still a relevant research theme [3]. The solution to enhance the batteries’ operational life or even eradicate their use through energy harvesting techniques has also turned into a topic of interest for both academia and industry. Several ambient sources have been proposed in the last decade to power these wireless nodes with energy captured from the environment [4].

On the other hand, among the batteryless IoT systems, sensor nodes based on RFID technology have become one of the most popular for short and medium-range applications [5]. RFID tags are remotely powered by the reader, but the reader must be near the tag as received power rapidly degrades with the distance. This is the most significant limitation of the RFID technology, especially when sensor capabilities are added to the tag to measure data in addition to RFID identification. Additionally, pure batteryless RFID tags cannot operate continuously if no RF source is available in the surroundings all the time. That makes a datalogging operation mode of the sensor difficult, since no stable supply energy is available. However, environmental energy harvesting can provide an endless energy source by collecting and storing energy from the ambient, thus having not only continuous energy for datalogging of sensor measurements, but also increasing the maximum communication range between reader and tag [6].

Depending on the application, different energy harvesting sources can substitute the traditional batteries with ambient energy transducers such as photovoltaic, piezoelectric, thermoelectric, and triboelectric modules, which convert, respectively, sunlight, vibrations, heat, or friction into electric power. Among them, solar energy provides the highest power density, producing enough power even under indoor light environments [7,8]. However, in indoor environments, the harvested power and output voltage are much lower. Therefore, in order to consider photovoltaic cells as a continuous energy source, its output needs to be adapted and managed by an electronic interface called Power Management Unit (PMU) that can supply the required voltage and power the corresponding load.

The main element of a PMU connected to a photovoltaic cell is the charge pump that raises the harvester output voltage up to the standard supply voltages of 1 V, 1.2 V, or 3.3 V required by different sensor nodes. Conventionally, charge pumps have been implemented with inductor-based architectures due to their high-efficiency [9,10,11]. However, to reach high performances, they rely on low-resistance and high-quality inductors, which are not available on standard CMOS technologies, if a fully integrated implementation is desired [11]. Nevertheless, monolithic converters built only with capacitors are gaining popularity for some applications because they can be fully integrated on ICs, suppressing expensive and voluminous off-chip LC components. Ref. [12] has a high efficiency in indoor lighting conditions, but it needs a high minimum input voltage, and it relies on an auxiliary charge pump to start-up. Ref. [13] integrates a photodiode on-chip as a solar cell for a full miniaturized system, although limiting its use only for outdoors. Ref. [14] works under indoor lights, but the power throughput is very little.

In this work, we present a self-sustaining energy-efficient monolithic PMU that includes a charge pump adapted to photovoltaic cells with the capability of charging a large supply capacitor and managing the stored energy efficiently to provide the required supply voltage and power to low energy consumption wireless sensor nodes such an RFID sensor tag. Previous works developed by the authors in [15] proved and validated the architecture of the charge pump with simulations. This paper continues the previous design by adding on-chip all the necessary blocks, implementing it for fabrication in silicon on the TSMC 180 nm technology and the post-characterization of the energy harvester.

The rest of the paper is organized as follows: Section 2 first introduces the complete harvester system characteristics and requirements, including the monocrystalline photovoltaic cell. Section 3 shows the circuit implementation and pays special attention to the monolithic charge pump. Section 4 explores the measurements and results of the fabricated IC, finishing with the conclusions in Section 5.

## 2. Proposed System

The proposed indoor solar energy harvester is capable of supplying power to a typical autonomous sensor node. Figure 1 shows the block diagram of the proposed autonomous sensor node. The different parts are described in the following subsections:

### 2.1. Load

As shown in Figure 1, the PMU acts as an interface between the energy source and the load and provides the required power to the microcontroller, the sensor and the RFID tag, which is in charge of the wireless communication with the reader. In order to optimize the energy, the MCU and the sensor are switched off when the transceiver is listening, waiting for the trigger command. Only the transceiver and the PMU remain active during the listening time.

Figure 2 shows the typical power consumption curve of a wireless sensor node load. During the active mode, the sensor collects it or manipulates the data requiring higher power consumption than that harvested by the energy generator. On the other hand, during the sleep mode periods, the sensor reduces its energy consumption by stopping non-critical functionalities switching off parts like the MCU core, sensor electronics, and other interface logic peripherals waiting for a stimulus to wake up. As a result, low-power duty cycled-based nodes have sleep mode periods at least one order of magnitude larger than the active mode. According to that, the power consumption of the sensor node can be classified as:**Power limited loads**: In these loads, the power consumption during the “Sleep Mode” is of the same order of magnitude as the one generated by the harvester. This usually happens when the transceiver continuously listens to incoming communications, leading to non-neglectable power consumption during that time. Therefore, in a power-limited load operation, it is key to maximizing the efficiency of the PMU by finding the maximum power point through the implementation of maximum power transfer point tracking modules (MPPT) that controls the charge pump and adapts it to the different operation scenarios over time;**Energy limited loads**: In this kind of load, the average power consumption during the “Active Mode” (measurement and data aresion) is at least one order of magnitude above the “Sleep Mode” consumption and average generated power. This way, during the sleep mode state, the system can store energy that is going to be consumed afterwards during the “Active Mode”. Therefore, the MPPT is not such a key aspect in these circuits as most of the energy is recovered and stored in an external capacitor with a minimal power consumption of the load, and the harvester can be considered to operate near the “open circuit” scenario in the stationary state.

In this work, we propose a PMU intended for the previously defined energy-limited loads working on the most restrictive scenario: Power consumption caused by the continuously listening low-power transceiver and a larger power consumption during the sensor and microcontroller activity. The power consumption curve is estimated considering the following parts in the system as a reference:

The transceiver Rocky100 RFID IC communicates with the reader or central node using backscattering techniques and is complemented with a battery-assisted semi-passive operation with −24 dBm of sensitivity that corresponds to 21 m of communication range [16]. The minimum energy needed to supply the chip on battery-assisted mode is 12.6 μW with a supply voltage range specification between 1 V and 3.3 V (minimum of 1.4 V to wake up) [17].

The chosen microcontroller is the *MSP430L092SPWR* low-power MCU, which would be supplied through the RFID IC only when a temperature request occurs and can operate with voltages as low as 0.9 V with a current consumption during the operation of 45 μA [18]. The MCU and the sensor are switched off during the sleep mode to optimize the power consumption.

The *TMP114* sensor communicates directly with the microcontroller and drains a current of 68 μA on average during the conversion and sensing time [19]. In order to build the power consumption curve, a 0.5 Hz measurement rate and 10 ms of sensing, conversion and processing time are considered.

Adding up these loads, the power consumption specifications from an energy point of view are a continuous power consumption of 12.6 μW during the sleep mode provoked by the RFID tag listening to the network and 19.8 μW on average during the active mode.

### 2.2. Energy Source

The selected energy source is a tiny monocrystalline photovoltaic panel *KXOB25-05X3F* with an active area of 1.42 cm^2^ [20]. The behaviour of the solar cell is characterized under different indoor illumination conditions, considering light intensities from 500 to 1000 lux and various light sources (Fluorescent, LED and Halogen) [21]. The photovoltaic generator is shown in Figure 3a, and the characterization results are plotted in Figure 3b.

The energy generator converts the environmental light into a continuous voltage that depends both on the external light intensity and the PMU power request following the mentioned power curve in Figure 3b measured with an LED light source.

When considering indoor energy harvesting systems, the light source type and its spectral power distribution should also be considered and not only the ambient illuminance level. Measurements of the peak output power of the aforementioned solar cell lighted under the most common indoor light sources (halogen, fluorescent and white LED) at different light intensities are shown in Table 1. The study reveals that halogen bulbs generate substantially more power than fluorescent or LED lights due to the wider power spectrum [22].

For this work, LED illumination is considered due to its popularity and efficiency. LED lighting technology is anticipated to be used by the majority of lighting installations by 2035, according to the US department of energy [23]. A typical standard indoor operation is considered as 680 lux with LED light [21], with a measured maximum power peak of 21.38 μW with the previously mentioned solar cell.

### 2.3. Power Management Unit

The main task of the PMU is to efficiently convert the harvested energy by the energy generator into a higher DC voltage that fulfills the load requirements. Moreover, the PMU enables/disables the RFID tag activity depending on the available energy, being able to supply high-power-demanding loads discontinuously with current vs. time profile as shown in Figure 2. The next section describes the different circuits that implement the PMU.

## 3. Architecture of the Proposed PMU

The proposed PMU architecture integrates the main DC-DC core with a floating negative input, which reduces the charge pump parasitic effects. The block diagram can be found in Figure 4. A digitally controlled oscillator (DCO) controls the operation point of the charge pump and provides the possibility to close the loop with an MPPT algorithm. Furthermore, an output energy monitor (EM) oversees energy availability on the storage capacitor and an overvoltage protection module (OVP) protects from unexpected increments of the input voltage coming from the energy generator. Finally, the output is driven by the load controller switch ($S_{LOAD}$) which activates or disables the load.

### 3.1. DC-DC Converter

The objective of the charge pump (CP) or DC-DC converter is to multiply by four the energy generator output voltage. It is composed of two identical charge pump modules ($CP_{1}$ and $CP_{2}$) with a fixed conversion ratio of 2×, which afterwards are summed by a conventional voltage adder obtaining the final 4× rate as depicted in Figure 5.

The 2× charge pump, described in Figure 6 and Table 2, is designed with a novel half-floating topology based on the proposed on [14], where the negative terminal of the energy generator is not hard-fixed to the circuit ground. This avoids the unproductive charging and discharging of the parasitic button-plate capacitors $C_{1}$ every clock cycle.

The proposed sub-block charge pump CP1/CP2 of Figure 5 operates in two cycles using the clock signals CLK and CLKN generated by the internal oscillator (DCO). As depicted in Figure 6a, during the first cycle $\phi_{1}$, when the CLK signal is high and CLKN is low, the voltage across $C_{1}$ is transferred to $V_{IN}$ by connecting the energy generator directly to the capacitor terminals while the output switch is opened. On the non-overlap intermediate phase (Figure 6b), both signals CLK and CLKN are set to low.

In the second cycle $\phi_{2}$ (Figure 6c), when the CLK signal is low and the CLKN is high, the circuit is rearranged to connect the energy generator in series with $C_{1}$, duplicating the input voltage to the output. At this point, the solar cell negative terminal is ground-referenced GND and the output switch $S_{LOAD}$ conducts.

The charge pump requires a non-overlapping clock signal, needed to avoid the shoot-through current loss during the switching process of the capacitor $C_{1}$ with ground (Figure 6) and $C_{4}$ over the switch $S_{1}$ and the transistor $M_{1}$ (Figure 5).

The multiplying factor of the complete CP is chosen as 4× to supply autonomous sensor nodes at the standard 1.8 V voltage level, when the output voltages of the solar cell are as small as 500 mV.

The external storage capacitor, connected to the $V_{out}$, smooths the pulsing nature of the switched 4× output and absorbs the load’s consumption peaks.

### 3.2. Digitally Controlled Oscillator

An internal relaxation oscillator provides the non-overlapping two-phase clock signal to operate the charge pump. The oscillator is supplied by the auxiliary 2× gain, $CP_{2}$ output (Vaux in Figure 4), with an unregulated voltage between 0.8 V and 1.2 V.

The architecture can be seen in the block diagram of Figure 7. A reference current (*Iref*) proportional to a reference voltage (*Vref*) is first generated Iref=k·Vref, where *k* is a constant proportional to the technology and internal resistor. Afterwards, a tunable charging current *Icap* proportional to *Iref* is generated following the equation Icap=n·Iref, where *n* is a digital tunable parameter varying from 1 to 256. *Icap* fills an integrated capacitor until it reaches the aforementioned Vref voltage level. When this value is reached, the comparator outputs go from low to high, and this signal is used to discharge the capacitor, returning the oscillator to the initial condition.

During the charging and discharging cycles, the comparator generates a train of short pulses, which are later converted by the flipflop to a regular 50%-duty cycle clock signal halving the frequency. The period is calculated from the general charging equation of a capacitor applying all the previous assumptions (Equation (Equation 1)). Finally, a non-overlapping module based on delays produces the required clock signal to operate the switches of the CP, producing the output frequency expressed by Equation (Equation 2): (1)$$T=\frac{V_{ref}\cdot C}{I_{ref}}=\frac{C}{k}$$
(2)$$f=\frac{k}{2C}$$

The selected clock topology, with an oscillation frequency independent from Vref (Equation (Equation 2)), ensures an output frequency independent from first order VDD-variations. The oscillator’s frequency can be tuned externally by increasing or reducing the charging current (*Icap*), resulting in changes in the oscillation frequency. The DCO-oscillator is controlled by the 8 bit digital signal FREQ. Measurements on the fabricated oscillator demonstrate a frequency range from 22 Hz to 1580 kHz. Figure 8 compares the measured and simulated frequency depending on the 255 steps of the 8 bit control signal. The output frequency in simulations is slightly lower than expected due to the extra parasitics on the paths, vias, and capacitors. This clock variation range is suitable to control the operation point of the CP and implement an MPPT to maximize its conversion efficiency.

### 3.3. Energy Monitor

The EM circuit continuously checks the storage capacitor voltage level and controls the $S_{LOAD}$ switch in Figure 4. The monitor output switching is designed with a fixed hysteresis (between the predefined threshold voltages $V_{1}$ and $V_{2}$) to fit the loads’ voltage specifications. Specifically, the $S_{LOAD}$ switch is closed (conducting) by the EM when the storage capacitor surpasses the predefined value of 2 V threshold, and it is opened (non-conducting) when the voltage drops under 1 V. In this way, the external capacitor value ($C_{ext}$) is appropriately chosen depending on the energy requirements of the load.

The EM supplies power to the load only when enough energy is available in the capacitor to complete the required task, avoiding unproductive circuit wakeups. Given a specific energy requirement by the wireless sensor node (ΔU) and the aforementioned voltage margins, the required capacitor value can be calculated by the following Equation (Equation 3):(3)$$\Delta U=\frac{1}{2}V_{2}^{2}C-\frac{1}{2}V_{1}^{2}C\overset{}{\to }C_{ext}=\frac{\Delta U}{1.5}.$$

In this work, the minimum capacitance applying Equation (Equation 3) for the consumption profile of Figure 2 for a consumption of 125 μW during the 10 ms of active time results in a capacitor of 0.555 μF. The storage capacitor is connected externally and is subjected to the energy consumption during the active period of the WSN on each application.

A size change in the external storage capacitor does not affect the harvester voltage output. Still, it has an impact on the charging time and, therefore, on the WSN activity rate. Considering an average generated power by the energy harvester of $P_{out}$, an idle power consumption in the WSN of $P_{sleep}$ during $t_{sleep}$ and an active energy consumption of $P_{active}$ during $t_{active}$, the maximum activity rate or duty cycle of the proposed WSN can be determined using the equilibrium of charges (Equation (Equation 4)), resulting in the following Equation (Equation 5). The calculated maximum duty cycle for this application following the previous consumption profile of Figure 2 with 125 μW of consumption during the active time, 12.6 μW during the inactive period and considering 0.5 Hz of sampling rate, Equation (Equation 5) results in a maximum duty cycle of 5.49%:(4)$$T_{cycles}\cdot P_{out}=t_{active}\cdot P_{active}+t_{sleep}\cdot P_{sleep}$$
(5)$$D_{active}=\frac{t_{active}}{T_{cycle}}=\frac{T_{cycle}\cdot P_{out}-t_{sleep}\cdot P_{sleep}}{P_{active}\cdot T_{cycle}}$$

### 3.4. Overvoltage Protection Circuit

The overvoltage protection circuit (OVP) protects all the circuitry from any sudden overvoltage. It is designed to sense any overvoltage condition at the output of the CP and drain the voltage excess to ground avoiding any harm to the entire chip or the load.

The overvoltage condition may occur when the input voltage exceeds the preset limits, for example, if the solar cell is exposed to outdoor illumination or transient high light peaks.

The absolute maximum voltage rating admissible in the chip is limited by the silicon technology: TSMC 180 nm general purpose, whose voltage limit yields about 3.3 V. Therefore, the upper cutoff voltage is designed slightly below 3.3 V. Measurements on the isolated OVP module show a leak current of 0.3 μA at 1.8 V and a cutoff voltage of 3.1 V, as it can be seen in Figure 9.

## 4. Implementation and Results

The proposed energy harvester has been implemented, paying special attention to the PMU, which has been designed, simulated, and fabricated in the TSMC 180 nm general purpose technology. Figure 10 shows the layout of the PMU, where the active area is 0.975 mm${}^{2}$ (650 μm × 1500 μm).

To integrate the flying capacitors of the charge pump, dual-layer metal–insulator–metal (MIM) on-chip capacitors are used. Due to the monolithic integration, the four on-chip capacitors fill most of the chip area, as it can be observed in Figure 10, where $C_{11}$ and $C_{12}$ correspond to the two 2× single sub-block charge pumps ($CP_{1}$ and $CP_{2}$), and $C_{3}$ and $C_{4}$ correspond to the voltage adder from Figure 5.

In order to characterize the whole harvester, the PMU has been encapsulated in a SOIC24 package and mounted on a PCB, as shown in Figure 11 for testing.

### 4.1. Test Set-Up

The energy harvester has been evaluated using the previously characterized *KXOB25-05X3F* solar cell as an energy generator. To ensure the right light calibration and measurements’ reliability and repeatability, the solar cell is fixed inside a dark box to avoid external light disturbances and illuminated exclusively with three white LEDs attached inside (Figure 12). This setup is employed only for testing purposes, while the energy harvester has the solar cell attached near the chip.

The light illuminance (lux) has been previously calibrated with a luxmeter and tabulated (Table 3) against the open circuit voltage $V_{OC}$ of the solar cell. Thus, the light intensity in lux can be regulated externally by adjusting the LED voltage in relationship to the tabulated open-circuit voltage.

To measure all the relevant characteristics of the energy harvester and cover the entire operating range, the three most significant scenarios are considered: first, the open-circuit operation demonstrates the highest achievable voltage when the storage capacitor is about to be fully recharged. The second test reveals the maximum efficiency point in a continuous mode that traditionally is the key feature of every energy harvester. Finally, the discontinuous mode validates how the energy monitor works with high power-demanding loads.

### 4.2. Open Circuit and Start Up

The PMU cold starts up automatically when light strikes the solar cell. Measurements indicate that the energy generator must provide a minimum voltage of 708 mV to start up the system. This means that the system is able to start in dark office environments with light intensity as low as 500 lux. The PMU initializes in the open circuit, and the load is only activated by the energy monitor when the external energy storage overcomes the 2 V threshold.

The voltage conversion efficiency (VCE), calculated through Equation (Equation 6), validates the conversion ratio of the main charge pump (4×). It relates the average output and input voltage with the theoretical conversion ratio of 4:(6)$$\eta_{VCE}=\frac{V_{out}}{CP_{convRatio}\cdot {\bar{V}}_{in}}$$

For this test, the clock frequency is swept from 22 to 1560 kHz to cover the full oscillator range. Figure 13 shows the output and input voltage of the PMU over the clock frequency, as well as the VCE. It can be seen that the energy harvester does not operate accordingly at low frequencies below 220 kHz due to the leak currents. The VCE’s best performance is achieved for a clock frequency between 400 and 500 kHZ, reaching the peak efficiency of 94.5% at 500 kHz. At high clock frequencies, from 1.4 MHz, the equivalent input impedance of the circuit varies, reducing the solar cell’s input voltage. This decrement in the input voltage causes a reduction in the output voltage.

### 4.3. PMU Performance

The overall significant figure-of-merit to characterize energy harvesters is the power conversion efficiency (PCE). The end-to-end measured efficiency is defined by the following Equation (Equation 7):(7)$$\eta_{PCE}=\frac{P_{out}}{P_{in}}=\frac{V_{load}^{2}}{R_{load}\cdot {\bar{V}}_{in}\cdot I_{in}}$$

The efficiency curve of the PMU is measured and presented in Figure 14, where the maximum peak results in 60.54% with a throughput power of 13.14 μW, while maintaining an output voltage greater than 1.9 V. The load at the maximum power peak is a 275 kΩ resistor, and the illumination intensity is 680 lux, the typical indoor office light. This way, the system meets the requirements for the continuous power supply of the proposed RFID sensor tag. The external storage capacitor is set to 47 μF, higher than the previously calculated minimum capacitance, in order to reduce the output ripple for the tests. During the maximum efficiency test, the frequency is optimized externally to achieve the peak for each load scenario.

The output power provided by the energy harvester is enough to continuously power the aforementioned wireless sensor node in Section 2.1 under indoor light conditions in sleep mode while the remnant power can be stored in the supply capacitor to provide energy to the sporadic operation of the microcontroller during the active mode. In order to increase the available current and therefore the external capacitor charging, it would also be possible to arrange two photocells in parallel as it is explained in [22] but at the expense of increasing the solar cell area and cost.

### 4.4. Discontinuous Mode

Thanks to the EM and the external storage capacitor, the PMU can provide, for a limited period of time, more power to the load than the continuously generated by the solar cell. On the following test represented in Figure 15, a load of 51 kΩ is connected to the harvester consuming a peak power of 63 μW. The storage capacitor used in this case is increased to 47 μF, to hold the voltage longer, during a 2.2 s for this case. The load is disconnected from the harvester automatically by the energy monitor through the $S_{LOAD}$ switch when the output voltage is below 1 V. Then, the external storage capacitor is recharged again and preventing the storage capacitor from a total discharge. When the output voltage rises above 2 V, the switch turns on the load again.

When using the minimum storage capacitor of 0.555 μF, calculated previously, the charging process of the storage capacitor is reduced to 140 ms, as depicted in Figure 16. Figure 16 demonstrates a full recovery from 1 V to 2 V of the 0.555 μF capacitor within 140 ms.

This example confirms the correct functionality of the energy harvester driving a discontinuous load profile, discussed previously in Section 2.1. The energy monitor activates and disconnects the load depending on the external capacitor voltage, and therefore the stored energy. In order to supply more energy-demanding loads, the supply capacitor should be increased, impacting its charging time and therefore the measuring rate, which is adjusted automatically by the EM operation.

The voltage drop on the $S_{LOAD}$ switch due to on-resistance averages 20 mV when driving 1.8 V to the load. On the other side, when the switch is not conducting, the off-resistance measures an average of 11.5 MΩ.

### 4.5. Comparison with the SoA

Finally, Table 4 compares the proposed circuit with other monolithic state-of-the-art solar energy harvesters with the same output power order of magnitude. The output power capability of this work under indoor light environments is high and in the same range as [12], which is considered the golden standard in terms of efficiency; however, it requires a much higher input voltage than the circuit proposed here and does not include any kind of energy monitor. On the other side, Refs. [13,24] integrate an energy monitor with a switch which separates the storage capacitor from the load, although in both cases reducing the efficiency below [12] and on the same order of magnitude of this work. However, both cases [13,24] are intended for outdoor use and are not optimized for a more restrictive indoor operation, as this work is. In terms of a silicon active area, this work presents a low active area comparing it with other circuits with the same node technology.

## 5. Conclusions

This work proposes an energy harvester, which features on-chip switched capacitors integrating all the auxiliary circuits monolithically, including the oscillator, the energy monitor, an overvoltage protection circuit, an output switch, and the charge pump among other test structures. It has been fabricated in standard TSMC general purpose 0.18 μm CMOS technology occupying a total active area of just 0.97 mm${}^{2}$.

Though this proposed harvester, the circuit has been validated with experimental measurements reaching a good efficiency under low indoor lighting conditions. The solar energy harvester is self-sustaining and cold starts without external biasing signals. The only source of power is the commercial solar cell KXOB25-05X3F with an active area of 1.42 cm${}^{2}$. The system is able to start up under indoor illuminations from 500 lux without any external kick-off or control signal. The measured end-to-end efficiency PCE achieved is 60.54% with a throughput power of 13.14 μW at 680 lux. The maximum output power under indoor light conditions at 1000 lux ranges up to 25 μW. When working on the open-circuit, the peak VCE reaches 94.5%. Finally, the integrated energy monitor gives the ability to supply any autonomous sensor nodes inside the 1 V and 2 V operation range with discontinuous operation independently of its power consumption during their activity by modifying the supply capacitor size and automatically adjusting its measuring rate. As far as authors know, the integration of this smart power management system together with a SoA energy harvester is new. The obtained results indicate that, with an improved PCE, an adjustable threshold in the discontinuous operation regions, and the use of new generation photocells such as the organic solar cells, the proposed topology will open new horizons in the development of batteryless autonomous sensor nodes.

## Figures and Tables

**Figure 1 sensors-22-07908-f001:**
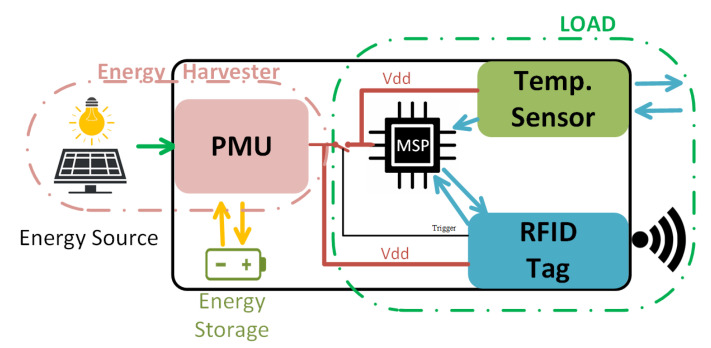
Proposed system block diagram.

**Figure 2 sensors-22-07908-f002:**
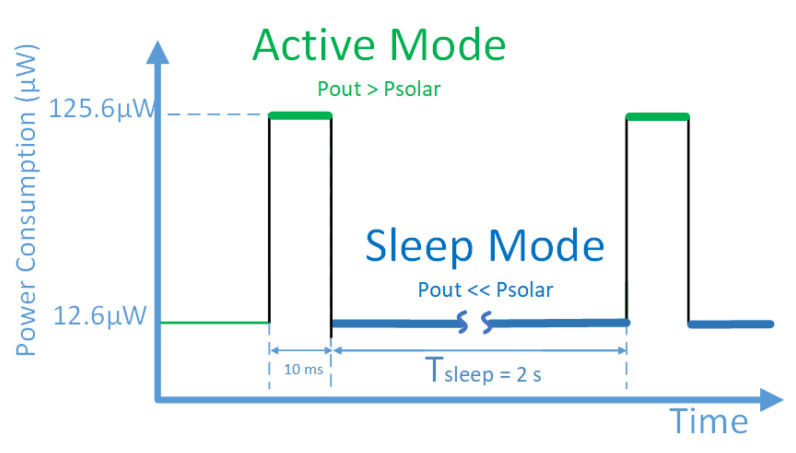
Power consumption pattern over time for the proposed wireless sensor node.

**Figure 3 sensors-22-07908-f003:**
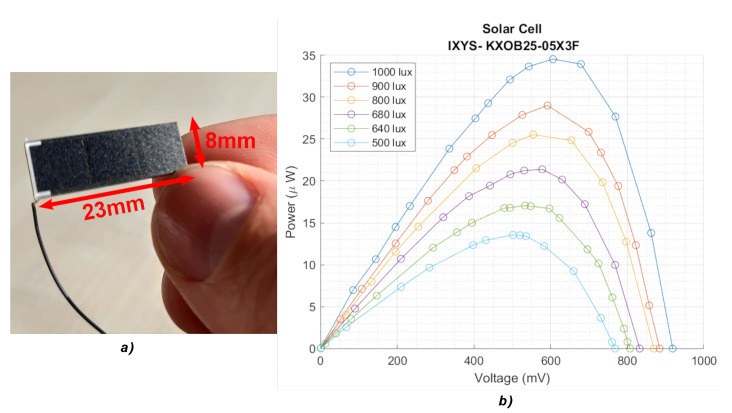
(**a**) Solar cell and dimensions; (**b**) output power from the solar cell vs. output voltage with LED light sources and different light intensities.

**Figure 4 sensors-22-07908-f004:**
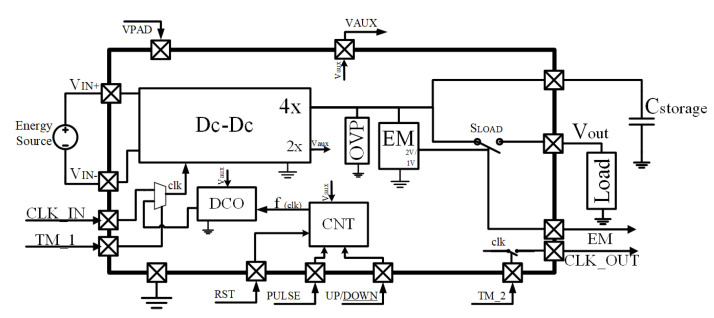
System architecture block diagram.

**Figure 5 sensors-22-07908-f005:**
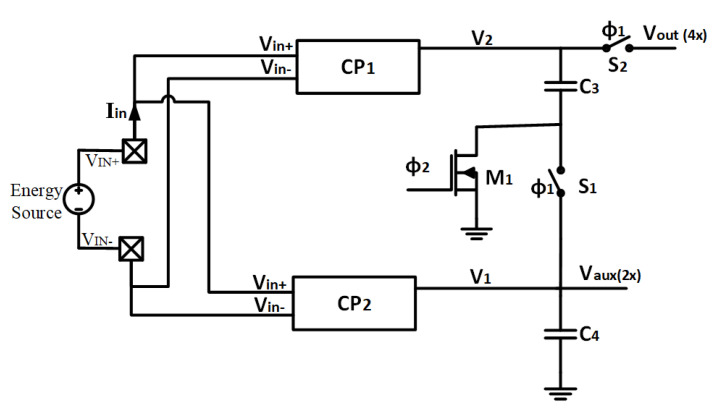
Main charge pump architecture.

**Figure 6 sensors-22-07908-f006:**
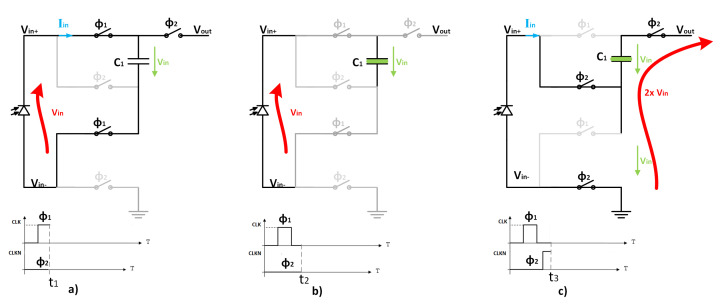
Operation cycles of the 2× charge pump sub-block. (**a**) first cycle with $\phi_{1}$ high; (**b**) overlap phase with $\phi_{1}$ and $\phi_{2}$ low; (**c**) second cycle with $\phi_{2}$ high.

**Figure 7 sensors-22-07908-f007:**
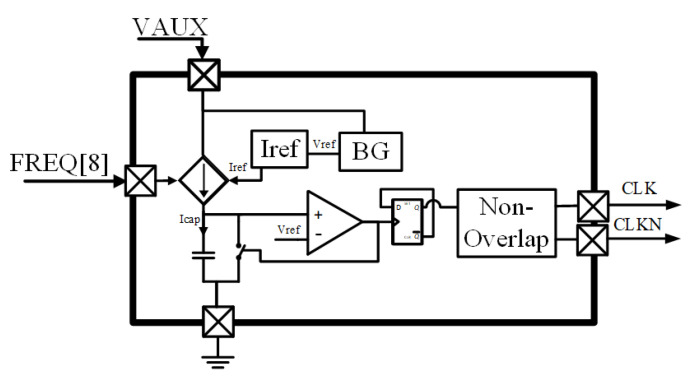
Oscillator block diagram.

**Figure 8 sensors-22-07908-f008:**
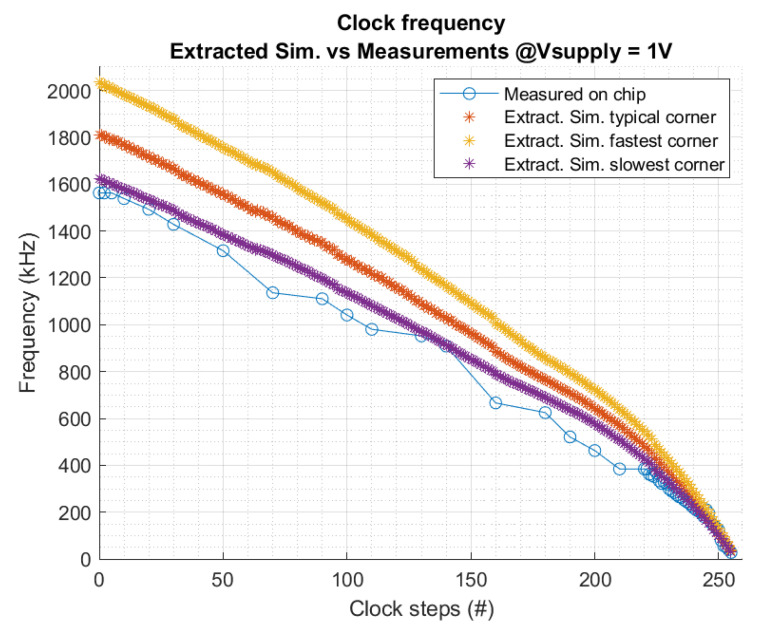
Comparison between fabricated and simulated output frequency.

**Figure 9 sensors-22-07908-f009:**
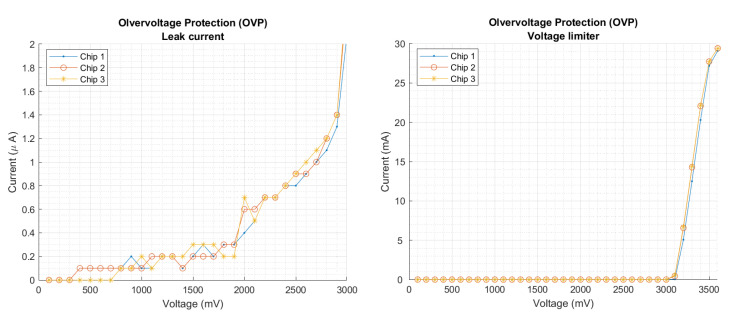
Characterization of the overvoltage protection module.

**Figure 10 sensors-22-07908-f010:**
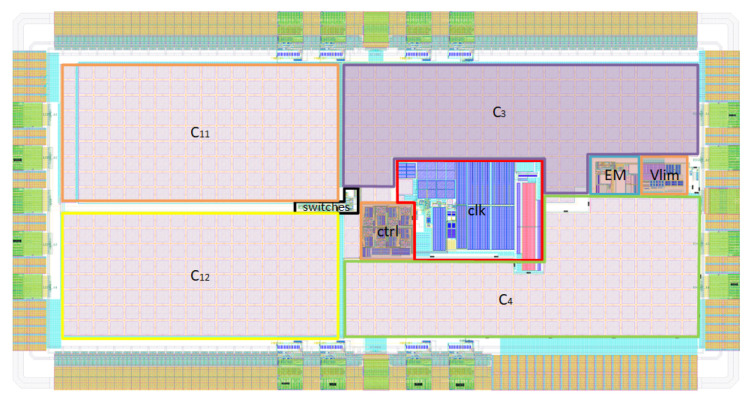
Layout micrograph.

**Figure 11 sensors-22-07908-f011:**
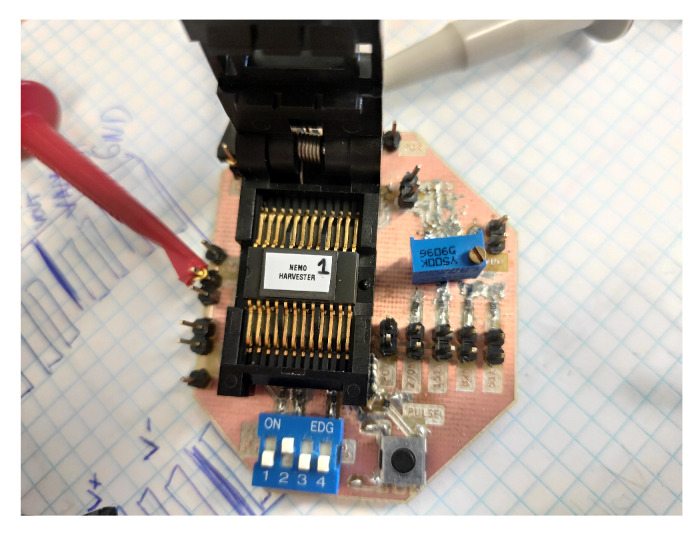
SOIC chip prototype and test PCB.

**Figure 12 sensors-22-07908-f012:**
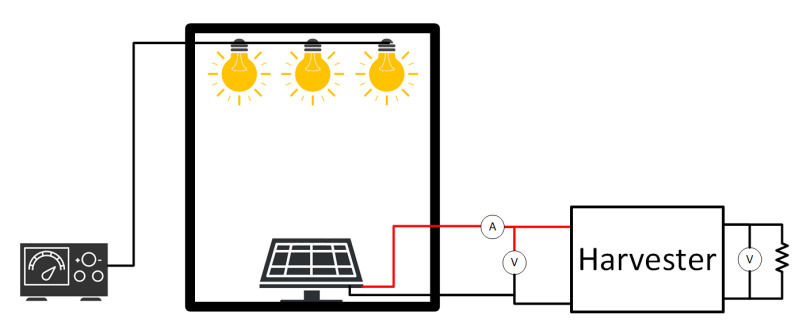
Test bench for the energy harvester with the light box.

**Figure 13 sensors-22-07908-f013:**
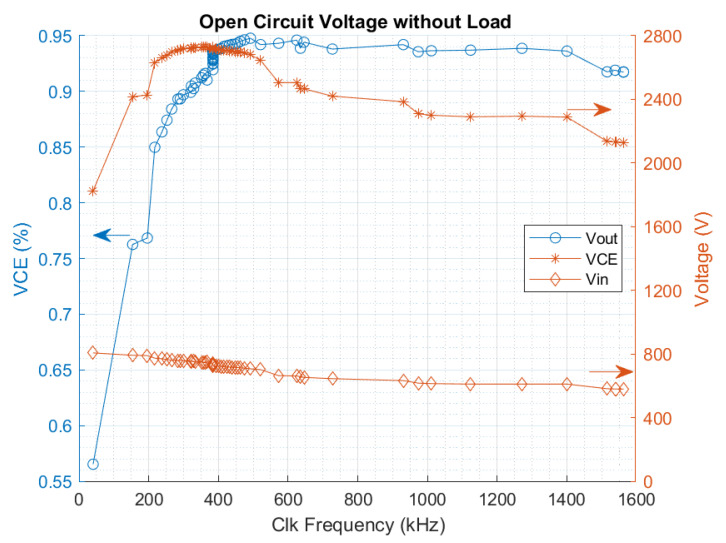
Measured output frequency of the oscillator.

**Figure 14 sensors-22-07908-f014:**
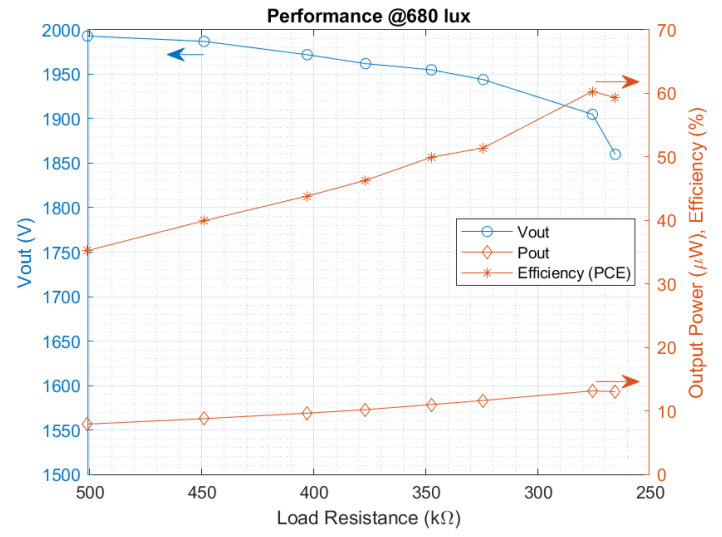
Measured overall conversion efficiency.

**Figure 15 sensors-22-07908-f015:**
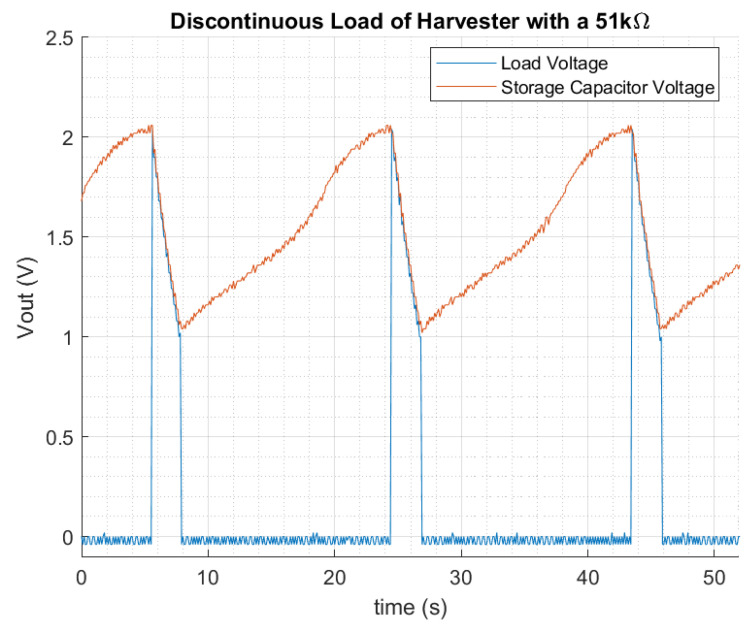
Discontinuous mode with a 51 kΩ equivalent to a peak of 63 μW.

**Figure 16 sensors-22-07908-f016:**
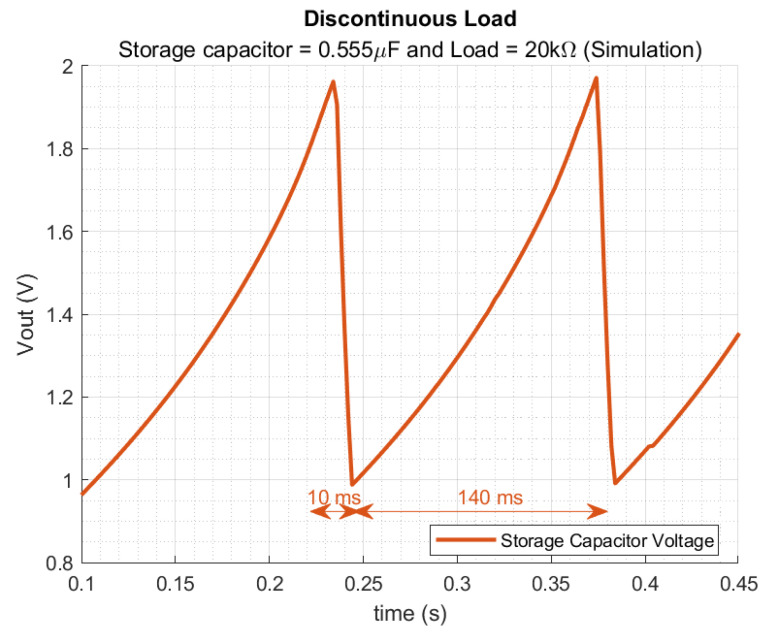
Discontinuous mode with a 0.555 μF storage capacitor and 20 kΩ.

**Table 1 sensors-22-07908-t001:** Peak power for different light sources and intensities.

Light Source\Light Intensity	500 lux	800 lux	1000 lux
Fluorescent	9.7 μW	16.55 μW	21.69 μW
LED	13.55 μW	25.53 μW	34.54 μW
Halogen	143 μW	223.6 μW	302.13 μW

**Table 2 sensors-22-07908-t002:** Circuit components’ dimensions.

		Width	Length	Multipliers
T-Gate	NMOS	1600 nm	300 nm	2×
	PMOS	220 nm	300 nm	2×
$C_{1-4}$	156 pF	30 μm	30 μm	88×

**Table 3 sensors-22-07908-t003:** Illumination correspondence and equivalence [21].

Illuminance (lux)	Open-Circuit Voltage (mV)	Equivalence
500	761	dark office
600	800	office light
1000	1023	bright lab illumination
3000	1205	very bright room
15,000	1508	outdoor indirect day-light

**Table 4 sensors-22-07908-t004:** Performance summary and comparison of a published fully integrated indoor solar energy harvester.

	This Work	[12] JSSC’15	[13] ICECS’19	[24] VLSI-DAT’20	[14] CICC’19	[25] JSSC’18
**Process**	180 nm	180 nm	180 nm	180 nm	180 nm	65 nm
**Input Range** **(mV)**	550–1200	1100–1500	400–600	250–480	450–800	350–1000
**Output Power** (μ**W)**	25 μW @1000 lx	21 μW @600 lx	0.5–5 μW @outdoor	0.5–50 μW @outdoor	2.5 nW–2.4 μW @300 lx	0.1–100 μW @outdoor
**Max. Efficiency** **@Pout** **@Vin** **@lux**	60.54% @12.25 μW @603 mV @680 lux	86.4% @12 μW @1250 mV * @420 lux	70% @ 1 μW * @400 mV	64.4% @30 μW	80.66% @1 μW @670 mV @280 lx	70% * @25 μW @500 mV
**Energy Monitor**	yes	no	yes	yes	no	no
**Area (mm${}^{2}$)**	0.97	2.25	2	1.15	1.95	0.54
**Solar Cell**	1.42 cm${}^{2}$	2.5 cm${}^{2}$	1 mm${}^{2}$ on-chip	-	3 cm × 0.58 cm	-

* Estimated from figures.

## Data Availability

Not applicable.

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
