# Peer review of "Power Management Unit for Solar Energy Harvester Assisted Batteryless Wireless Sensor Node"

_sensors, 2022, doi:10.3390/s22207908_

Round 1

Reviewer 1 Report

The article deals with an attractive topic. The content is well structured. The presentation of the theoretical as well as the experimental work is solid. The comparison with related solutions is illustrative.

When checking the references, I noticed that you have articles 23, 24, 25, and 26 in the list, which are not cited in work itself.

I also notice that you failed to reference your previous works: "Switched Capacitors Charge Pump with Half Floating Topology for a High-Efficiency Solar Energy System Harvester" (https://doi.org/10.1109/DCIS51330.2020). It's a related topic, or your current article is an upgrade of the previous one.

It would be correct to cite your previous work as a reference and clearly define the differences and innovations in the article concerning the previous article. It would also be good to elaborate on the essence of the scientific contribution of your work in the conclusion itself.

Author Response

Dear reviewer, thanks for your feedback and valuable comments. Please find below the answers to your specific comments:

  1. References 23, 24, 25, and 26 have been deleted from the reference section. They were there by mistake. Thanks for noticing
  2. A new paragraph about the previous related work has been added (line 70) remarking on differences and innovations between both: previous work was at the simulation level whereas this work improves the architecture and achieves the implementation and characterization stage of the proposed circuit. Measurement results are reported.
  3. Last, the conclusion section has been improved by remarking the essence of the scientific contribution including the next phrase: As far as authors know, the integration of this smart power management system together with a SoA energy harvester is new. The obtained results indicate that with an improved PCE, an adjustable threshold in the discontinuous operation regions and the use of new-generation photocells such as organic solar cells, the proposed topology will open new horizons in the development of batteryless autonomous sensor nodes.

Once again thanks for your comments.

Reviewer 2 Report

The authors describe a PMU for the batteryless wireless sensor node by using the power from solar energy harvesting. This is a creative idea for the sensor node to use a solar cell system to replace the battery, especially it works under weak environment light. I would suggest to publish after a minor revision as following:

1. In this manuscript, author introduce many abbreviations of the functional unit. I suggest author should use the full name and abbreviation when they appear first time. such as the first line in abstract “PMU (Power Management Unit) can be replaced by “Power Management Unit (PMU)” … Even I can find the list of abbreviations at the end of the manuscript, I think with the full name in the manuscript will make it more readable.

2. Do you have the reference lifetime of the solar cell (KXOB25-05X3F) used in this manuscript? As you mentioned in the intro, the battery may need to replace in 5 or 10 years. I wonder if the solar cell can work perfectly in such long period.  Or is it easier to replace than a built-in battery? Just curious, could you try organic solar cells if it is easier to replace, even the stability of organic solar cell is much less than the silicon cells.

As the test, I found the solar cell and sensor node are separated two part as the Figure 12 shown. Just curious if it is the same as a whole device. The cells need light to support the sensor. It should be on the surface of the device, and no cover on it, right?

Author Response

Dear reviewer, thanks for your feedback and valuable comments. Please find below the answers to your specific comments:

  1. The abbreviations have been checked and updated.
  2. The specific lifetime of the implemented solar cell is not revealed by the manufacturer. It is just claimed as “Long life and stable output”. Despite this, monocrystalline solar cells are designed to last between 25 - 40 years because they have fewer potential failure points due to the fact they are built with one single crystal. This feature is added in the introduction on line 25 remarking the advantage of them over built-in batteries.
  3. Organic solar cells are a nice alternative to traditional silicon cells, but the technology is still developing, and the efficiency and lifespan are far from traditional silicon photovoltaic cells. Therefore, they are not considered for this case. However, they are mentioned in the conclusion as possible future work.
  4. Respecting the test setup, the light box is used only for testing purposes. Later on, the normal use, the solar cell is attached near the chip. This is clarified in the corresponding section.

Once again thanks for your comments.

Reviewer 3 Report

The paper is well written, the structure is fine, the results and analysis are in details. Just several minor issues, that should be revised.

1.        Line 36, check here …supply energy i.

2.        The introduction should be improved, currently, the introduction section provides the general topic introduction, which is fine. However, the state-of-the-art review and analysis are missing, undermining the justification of the novelty and relevance of the current paper goals.

3.        The limitations and future works should be reported in the Conclusion.

Author Response

Dear reviewer, thanks for your feedback and valuable comments. Please find below the answers to your specific comments:

  1. As you commented, the errata of line 36 is now corrected. Thanks for noticing it.
  2. The state-of-the-art paragraph has been extended in the introduction (lines 53 to 65) with more reviews of the state-of-the-art solutions and remarking on the improvements of the proposed one.
  3. The limitations and future works have been reported in the conclusion section by adding the following phrase: The obtained results indicate that with an improved PCE, an adjustable threshold in the discontinuous operation regions and the use of new generation photocells such as organic solar cells, the proposed topology will open new horizons in the development of batteryless autonomous sensor nodes.

Once again thanks for your comments.